# A recurrent neural network for network-based intrusion detection

Abraham Berhanu[1], Sabarathinam Chockalingam[2], Jemal Abawajy[3,4],
Shegaw Anagaw Mengiste[5], Shabbab Ali Algamdi[6] and Dereje Ashenafi[1]

[1] Department of Electrical and Computer Engineering, Addis Ababa Science and Technology University, Addis Ababa, Ethiopia
[2] Department of Risk and Security, Institute for Energy Technology, Halden, Norway
[3] Faculty of Science, Engineering and Built Environment, Deakin University, Burwood, Australia
[4] Faculty of Computer Science, University of Brawijaya, Malang, Indonesia
[5] School of Business, University of South-Eastern Norway, Notodden, Norway
[6] Department of Software Engineering, College of Computer Science and Engineering, Prince Sattam bin Abdulaziz University, Al Kharj, Saudi Arabia



## ABSTRACT

Intrusion Detection Systems (IDS) play a vital role in monitoring and preventing unauthorized access within critical network infrastructures. However, current IDS approaches often suffer from limitations in detection accuracy, scalability, and adaptability to evolving threats. To address these challenges, this study proposes an optimized Long Short-Term Memory Recurrent Neural Network (LSTM-RNN) model for network-based intrusion detection. The model is specifically designed to capture temporal dependencies in sequential network traffic data, enabling accurate classification of benign and malicious activity while maintaining computational efficiency. The proposed model underwent rigorous validation to assess both its detection accuracy and generalizability. This was achieved by utilizing two distinct datasets that encompass realistic network traffic and a wide array of attack scenarios, including Denial of Service (DoS), Probe, User-to-Root (U2R), and Remote-to-Local (R2L) intrusions. The performance of the model is compared with several other recent models. The proposed model achieved high classification results among all the compared models, with an accuracy of 99.94%, precision of 99.98%, recall of 99.93%, and an F1-score of 99.95%. These outstanding results underscore the model's robust capability to accurately identify and flag malicious activities within complex network environments. The model's generalization power was similarly affirmed on a second independent dataset. The model maintained strong performance with an accuracy of 99.42%, precision of 99.20%, recall of 99.69%, and an F1-score of 99.45%. The successful demonstration of both high detection accuracy and strong generalization capability highlights the practical utility and inherent reliability of our proposed model for real-world threat detection.

## INTRODUCTION

Pervasive digitalization, driven by new technologies such as the Internet of Things (IoT) and edge computing, has brought remarkable transformations across various domains like education, healthcare, business, and entertainment (*Khan, 2021*; *Abawajy et al., 2018*;

Corresponding author
Jemal Abawajy,
jemal.abawajy@deakin.edu.au

*Hassan et al., 2020*). This transformation is accompanied by a growing number of interconnected devices continuously sharing information, particularly within critical systems like power grids and medical institutions. This increased connectivity significantly expands the attack surface area, rendering these critical infrastructures highly susceptible to cyber-attacks, which are increasingly prevalent (*Lallie et al., 2021*). Such successful attacks pose significant risks not only to individuals and organizations but also have wide-ranging consequences for society as a whole. They can result in substantial financial losses, compromised sensitive information, disrupted critical infrastructure, and even jeopardized national security. To mitigate these vulnerabilities, several conventional security mechanisms such as firewalls, encryption, and access control (*e.g.*, end-user authentication and authorizations) are implemented. However, despite their individual effectiveness, these methods often lack comprehensive packet analysis, which poses a significant challenge in detecting all types of attacks (*Hacılar et al., 2024*). This limitation highlights the critical need for advanced security solutions capable of detailed traffic inspection and real-time threat identification.

An Intrusion Detection System (IDS) is a critical first-line defence in cybersecurity. Its primary function is to vigilantly identify and monitor unauthorized activities, intrusions, or signs of cyber-attacks within computer and communication networks, with the goal of safeguarding against malicious threats and preserving essential security services (*Akgun, Hizal & Cavusoglu, 2022*). IDSs are primarily categorized by their deployment approach into Network Intrusion Detection Systems (NIDS) and Host Intrusion Detection Systems (HIDS). NIDS focus on analysing network traffic flows or traces, typically sourced from devices like firewalls, routers, or switches. In contrast, HIDS monitors host-based data sources, such as system logs or Central Processing Unit (CPU) load, to detect anomalies or malicious activity within individual systems. Hybrid IDS combines elements of both NIDS and HIDS for comprehensive intrusion detection. Beyond deployment, IDSs can also be functionally classified based on their attack detection procedure into signature-based and anomaly-based systems (*Yang et al., 2023*). Signature-based IDSs (SIDS) identify known malicious behaviour by comparing observed activity to predefined attack signatures. Conversely, Anomaly-based IDSs (AIDS) detect deviations from established normal system behaviour, flagging these unusual events as potential threats. These systems provide early detection of potential cyber-attacks, allowing security teams to respond promptly and protect the network and systems from unauthorized access or malicious activities (*Aldarwbi, Lashkari & Ghorbani, 2022*).

The escalating network traffic coupled with the persistent evolution of cyber threats underscores the paramount need for highly efficient NIDS to ensure robust network security (*Mynuddin et al., 2024*). Despite extensive research in the field, current NIDS methods frequently struggle with high false alert rates (*Liu, Gao & Hu, 2021*) and low detection rates (*Wang et al., 2021*). Traditional rule-based or signature-based IDSs, in particular, prove largely ineffective against emerging or unknown threats (*Kandhro et al., 2023*; *Park et al., 2021*; *Jayalaxmi et al., 2022*; *Halbouni et al., 2022*; *Ncir, HajKacem & Alattas, 2024*). Consequently, there is an urgent demand for more effective and efficient NIDS models capable of accurately identifying novel and sophisticated network intrusions.

To this end, we proposed an optimized Long-Short Term Memory Recurrent Neural Network (LSTM-RNN) based IDS that accurately detects and classifies network intrusions by exploiting the temporal and sequential patterns inherent in network traffic data. This work significantly contributes to the field of network intrusion detection by providing a model that not only improves accuracy and handles complex dependencies but also facilitates effective real-time detection of network intrusions. Major contributions of this study are as follows:

- Identified novel features that can enhance the model's ability to detect network intrusions.
- Proposed an optimized LSTM-RNN model's architecture capable of accurately modeling temporal dependencies in network traffic for real-time intrusion detection.
- Rigorously evaluated the proposed optimized LSTM-RNN model using widely used datasets and standard evaluation metrics, demonstrating its robust performance/effectiveness.

The rest of this article is organized as follows: 'Related Work' reviews existing LSTM-RNN based network intrusion detection approaches. 'Background' present essential background on LSTM-RNN and the datasets used in this study. 'Methodology and Proposed Model' of this article present methodology and proposed model. 'Results' provides analysis of the experiment conducted using our proposed model followed by 'Discussion'. Finally, 'Conclusions and Future Work Directions' provides key conclusions and potential future work directions.

## RELATED WORK

This section provides a comprehensive review of existing literature pertaining to NIDS, establishing the foundational context for our proposed methodology.

Deep Learning Approaches in Network Intrusion Detection: Deep learning has emerged as a pivotal approach in IDS, significantly enhancing their capability to detect and respond to diverse cyber threats (*Alshingiti et al., 2023*). Unlike traditional rule-based or signature-based IDS, which often struggle with novel and unknown threats, deep learning, particularly anomaly-based methods, leverages baseline normal behaviour to identify deviations, offering a promising avenue for detecting sophisticated and zero-day attacks. Given the increasing complexity and sophistication of contemporary cyber threats, researchers have actively explored various deep learning techniques, with a particular emphasis on models adept at processing sequential network traffic data. Long Short-Term Memory (LSTM) networks, a type of Recurrent Neural Network (RNN), are widely adopted for this purpose due to their inherent ability to model temporal dependencies.

Hybrid and Optimized LSTM Architectures for Enhanced Detection: Numerous studies have explored advanced LSTM-based architectures, often integrating them with other deep learning components or optimization techniques to enhance detection performance. For instance, several hybrid models combine LSTMs with

Convolutional Neural Networks (CNNs) for robust spatial and temporal feature extraction. *Kanna & Santhi (2022)* developed a Black Widow Optimized Convolutional-LSTM network, achieving 98.25% accuracy on CIC-IDS-2018. Similarly, *Hnamte & Hussain (2023)* reported impressive 100% accuracy on CIC-IDS-2018 with a Deep CNN-BiLSTM architecture, leveraging CNN for feature extraction and BiLSTMs for sequence prediction. *Halbouni et al. (2022)* also combined CNNs and LSTMs, incorporating batch normalization and dropout, attaining strong accuracies (*e.g.*, 99.64% on CIC-IDS-2017). Beyond architectural hybridization, researchers have focused on optimizing LSTM performance. *Manivannan & Senthilkumar (2025)* proposed an Adaptive RNN with a bio-inspired Fox Optimizer for improved accuracy (98.7% on NSL-KDD and CIC-IDS-2017) and faster convergence. *Salim & Lahcen (2023)* introduced a bidirectional LSTM model with a novel symmetric logarithmic loss function, achieving 99.92% accuracy on CIC-IDS-2018. *Donkol et al. (2023)* also enhanced LSTM-RNNs using likely point particle swarm optimization to mitigate gradient-clipping, demonstrating higher performance on CIC-IDS-2018 and CIC-IDS-2017 datasets.

Addressing data complexity and generalization challenges: Another significant area of research focuses on improving the generalization capabilities of LSTM-based IDSs and handling complex data characteristics. *Abdulmajeed & Husien (2022)* explored dataset amalgamation with a hybrid CNN-LSTM to enhance generalization, particularly during cross-dataset evaluation on CIC-IDS-2018 and CIC-IDS-2017. *Huang (2021)* introduced a spatiotemporal LSTM model with an unsupervised encoder for extracting spatial characteristics, leading to improved intrusion detection on NSL-KDD. *Hnamte et al. (2023)* utilized a two-stage LSTM and Auto-Encoder (AE) approach, reporting high multi-class detection accuracies on CIC-IDS-2017 (99.99%) and CIC-IDS-2018 (99.10%). Furthermore, *Han & Pak (2023)* proposed a hierarchical LSTM-based IDS, which achieved high detection rates (*e.g.*, 99.70% on CIC-IDS-2017) by considering information from the entire packet, overcoming limitations of existing methods. *Rizvi et al. (2023)* also contributed a 1D-Dilated Causal Neural Network, demonstrating high accuracy (99.98% on CIC-IDS-2018) with a simplified architecture for binary classification.

Research gaps and motivation: Overall, the literature showcases significant advancements in deep learning-based intrusion detection, particularly with LSTM-driven models and their hybridizations. These studies, often utilizing benchmark datasets like CIC-IDS-2018, demonstrate promising accuracy rates in cyber threat detection. However, despite these promising results, challenges persist regarding high false alert rates and the robust detection of truly novel or sophisticated threats in dynamic, real-world network environments (*Liu, Gao & Hu, 2021*; *Wang et al., 2021*; *Kandhro et al., 2023*; *Park et al., 2021*; *Jayalaxmi et al., 2022*; *Halbouni et al., 2022*; *Ncir, HajKacem & Alattas, 2024*). Thus, while these deep learning approaches demonstrate considerable potential, the literature still presents a notable gap in comprehensively addressing the combined demands for scalability, high accuracy, and real-time efficiency required for robust and effective deployment of NIDS. Overall, these studies propose innovative LSTM-based approaches and models for IDS, as summarized in Table 1. While various models exhibit promising

**Table 1 Comparative analysis of LSTM based network IDS.**

| Reference | Model | Datasets | Contribution | Evaluation metrics | Challenge addressed |
|---|---|---|---|---|---|
| Manivannan & Senthilkumar (2025) | ARNN-FOX | NSL-KDD and CSE-CIC-IDS2017. | Enhance detection of zero-day attacks, DDoS and malware | Accuracy, Precision, Recall, F1-score | Improving intrusion detection accuracy while reducing false alarms in dynamic networks. |
| Hnamte & Hussain (2023) | LSTM-AE | CICIDS2017 and CSE-CICDIS2018. | Enhances the understanding and effectiveness of IDS. | Accuracy, loss, recall, precision, and F-measure. | Dynamic nature of attacks and dataset updates and benchmarking. |
| Salim & Lahcen (2023) | Bi-LSTM | NSLKDD, CSE-CIC-IDS2018, and UNSW-NB15. | Enhancing the accuracy of network IDS. | Accuracy, precision, recall, and F1-score. | Address the limitations of existing methods in terms of long-term viability, practicality, adaptive performance, and detection precision. |
| Abdulmajeed & Husien (2022) | CNNBiLSTM | CICIDS2018 and Edge_IIoT. | Architecture provides more accurate predictions compared to single models. | Accuracy and confusion matrices. | Accurately detects irregular attack patterns. |
| Han & Pak (2023) | Hierarchical LSTM | ISCXID2012 and CI-CIDS2017. | Capture fine-grained features from individual packets and session-level patterns and achieve higher detection accuracy. | Detection rate, F1-score, and accuracy. | Addresses the challenge of using all packet data for intrusion detection without increasing the model's size excessively. |
| Rizvi et al. (2023) | 1D-DCNN | CICIDS2017 and CSE-CICDIS2018. | Improves existing IDS approaches, in terms of computational resources and processing time. | Accuracy, precision, and detection rates. | Developing an intrusion detection system suitable for resource-constrained environments. |
| Donkol et al. (2023) | ELSTM with LPPSO optimization. | NSL-KDD CICIDS2017, CSE-CIC-IDS2018. | Overcome the drawbacks of existing methods that make them inefficient against new/distinct attacks. | Accuracy, precision, recall, and error rate. | Addresses issues of gradient vanishing, generalization, and overfitting. |

accuracy rates across datasets, aiming to address specific challenges such as detection precision or architectural efficiency, they often lack scalability and generalization needed for practical deployment.

# BACKGROUND

To provide comprehensive context for our proposed methodology, this section begins by elucidating the foundational structure of LSTM networks. Following this, we present a detailed description of the datasets, and the performance evaluation metrics employed to rigorously assess our model's performance.

## Long short term memory

LSTM is a variant of RNN architecture introduced by Hochreiter and Schmidhuber in 1997 and has gained significant popularity in the field of deep learning (Borisenko et al., 2021). RNNs belong to a category of neural network architectures explicitly designed for

processing sequential data. RNNs are characterized by their recurrent connections, which allow information to be passed from one step of the sequence to the next. They maintain an internal hidden state that acts as a memory, enabling them to capture and utilize information from previous elements in the sequence when making predictions at each time step. The present output of an RNN sequence as depicted in Fig. 1 is influenced by both the current input and the previous output. It serves as a memory for information at the forefront of the network. When traversing through time, it can be conceptualized as a limited multilayer deep learning network. In RNN, each hidden state layer's fundamental role is to retain data and incorporate new information during each iteration, facilitating the downward passage of information (*Ashraf et al., 2020*).

Despite their strengths, RNNs have some shortcomings that can limit their performance and hinder their capability to capture and maintain long-term dependencies within sequential data. One of the major challenges with RNNs is the vanishing gradient problem. When training deep RNNs, the gradients that flow backward through time tend to diminish exponentially, making it difficult for the network to capture and propagate information over long sequences. This limits the RNN's ability to learn from distant past time steps and this could lead to the loss of relevant information. The opposite issue to the vanishing gradient problem is the exploding gradient problem. In some cases, gradients can grow exponentially during backpropagation, leading to unstable training and convergence issues. LSTM, an enhanced recurrent neural network, primarily addresses the limitation of RNNs in handling long-distance dependencies effectively (*Sohi, Seifert & Ganji, 2021*).

The training process of the LSTM model employs the time backpropagation algorithm. Within the LSTM model, the forward propagation algorithm is achieved through the internal self-recurrent state update method, computing the output value of each individual module. Subsequently, error terms for each LSTM are calculated in reverse from both the time and network layers. This enables the computation of weight gradients for the error term (*Xiong, Luo & Li, 2023*). Ultimately, the weight updates are performed using a gradient-based optimization algorithm.

The LSTM design incorporates an internal self-recurrent mechanism where the weights in this recurrence are context-dependent. This enables the neural network to selectively forget old states while ensuring a consistent gradient flow. Consequently, LSTM is not merely a straightforward nonlinear model with element-wise operations post-input and recurrent unit transformations. Instead, it is a sophisticated nonlinear system with a gating unit system comprising numerous parameters and controlling information flow (*Yang et al., 2022*). The LSTM gating unit system comprises three components: the forgetting gate, update/input gate, and output gate, as illustrated in Fig. 2.

In the LSTM framework, $x_t$ represents the current input, $h_t$, and $h_{t-1}$ denote the current and previous hidden states, $f_t$ signifies the forgetting gate, $g_t$ indicates the input gate, $q_t$ represents the output gate, and U, W, and B represent weight parameters in the self-recurrent aspect. Additionally, sigma() signifies the sigmoid activation function, while tanh denotes the hyperbolic tangent function.

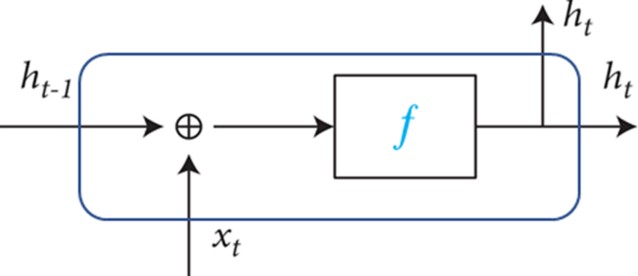

**Figure 1 RNN illustrating the internal structure of the hidden layer.**

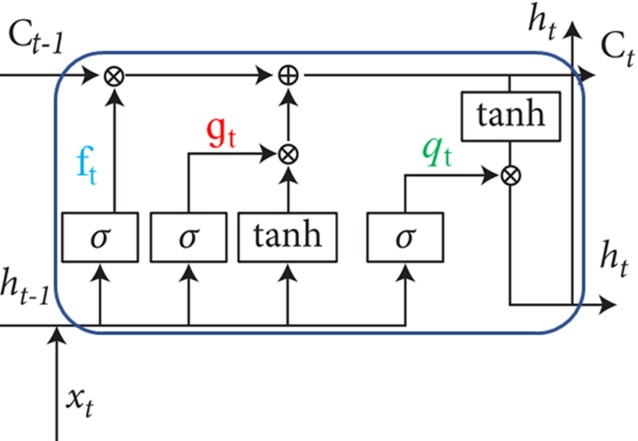

**Figure 2 Structure of LSTM gates.**               

The forget gate layer examines input data and information received from the preceding hidden layer, determining which information to discard from the cell state. This decision is made by combining the current input $X_t$ with the previous hidden state $X_{t-1}$. The output is then generated using the sigmoid function and multiplied by the previous cell state $C_{t-1}$. If the sigmoid function yields a value of 1, the information is retained; if it is 0, the information is deleted. It is calculated as:

$$f_t = \sigma\left(b_f + \sum U_f x_t + \sum W_f h_{t-1}\right). \tag{1}$$

The Update/Input gate layer determines the information that LSTM will store in the cell state $C_t$. Initially, the input gate layer uses a sigmoid function to decide which information to update. Following this, a tanh layer suggests a new vector to augment the cell state. Subsequently, the LSTM updates the cell state by discarding the designated forgettable information and replacing it with the values from the new vector. It is calculated as:

$$g_t = \sigma\left(b_g + \sum U_g x_t + \sum W_g h_{t-1}\right). \tag{2}$$

$$C_t = f_t C_{t-1} + tanh\left(b + \sum U_i x_t + \sum W_i h_{t-1}\right). \qquad (3)$$

The Output Layer determines the output by employing a sigmoid function, which dictates the portion of the cell that LSTM will output. The current cell state $C_t$ undergoes a tanh activation function, and the result is multiplied by the output of the sigmoid activation function of the output gate. This process generates the current hidden state $h_t$, representing the output of the LSTM network. The output is constrained between −1 and 1, ensuring that only the chosen information is transmitted to the next neuron. It is calculated as:

$$q_t = \sigma\left(b_o + \sum U_o x_t + \sum W_o h_{t-1}\right). \qquad (4)$$

$$h_t = tanh(C_t) q_t. \qquad (5)$$

During the training process, neurons are randomly dropped in each epoch using this technique. This is essential for preventing overfitting in deep neural networks, where the network may learn too well, hindering its ability to generalize to new samples. The final layer is fully connected, with each neuron connected to all neurons in the layer beneath it. This layer is responsible for classification, accomplished through the Softmax activation function (*Xiong, Luo & Li, 2023*). The input data is transformed into a one-dimensional layer to classify it into the appropriate class, assigning output probabilities. The output from this layer serves as the final output.

LSTM-RNNs were selected for intrusion detection in this study due to their effectiveness in handling sequential data. In computer network systems, data often appears in sequences, such as network traffic logs or system activity logs. LSTM-RNNs are a suitable choice for this task because they excel at retaining and learning from long-term dependencies within these sequential patterns (*Kocher & Kumar, 2021*).

## CIC-IDS-2018 dataset

The CIC-IDS-2018 dataset is a cybersecurity dataset that is commonly used for evaluating IDSs. It was created by the Canadian Institute for Cybersecurity (CIC) at the University of New Brunswick. The datasets were gathered over ten days in the year 2018. Available at https://www.unb.ca/cic/datasets/ids-2018.html (accessed February 22, 2024). The dataset was designed to provide a realistic and comprehensive set of network traffic data for evaluating IDS techniques. It contains various types of network attacks and normal activities, allowing researchers to develop and test intrusion detection algorithms (*Leevy & Khoshgoftaar, 2020*).

The CIC-IDS-2018 dataset, encompassing network traffic data portraying both normal and malicious activities, forms the basis for training and assessing the model. Preprocessing steps are applied to address duplicates, missing values, and feature scaling. This dataset comprises diverse network attacks, encompassing benign (or "normal") network traffic and harmful (or "malicious") network activity including brute force attack, botnet, Denial of Service (DoS)/Distributed Denial of Service (DDoS) attack, web attack, and infiltration attack.

Brute Force Attack involves a trial-and-error method commonly employed for password cracking and discovering concealed content or pages, relying on the chance of success. A Botnet consists of internet-connected devices utilized by its owner to perform various tasks, including data theft, spam delivery, accessing restricted areas, and facilitating access to the attacker. A DoS attack seeks to temporarily disable a system or network resource by overwhelming it with excessive requests, impeding its functions. DDoS attack occurs when a victim's resources are overwhelmed by numerous systems, flooding the targeted system with network traffic and resulting in a DoS, often orchestrated by a botnet.

Web attacks, constantly evolving, exploit vulnerabilities like SQL Injection and Cross Site Scripting (XSS), showcasing risks such as Brute Force over HTTP, where an attacker attempts multiple passwords to uncover an administrator's password. Infiltration attacks typically follow successful reconnaissance and penetration, involving privilege escalation for the attacker to navigate the compromised system, identify targeted systems, devices, and potentially gain higher access levels.

The selection of the CIC-IDS-2018 dataset for this research is attributed to its comprehensive coverage, incorporating a wide array of cyber threats including DoS, DDoS, and brute force attacks. This dataset offers a realistic portrayal of diverse attack scenarios encountered in real-world environments. Its substantial scale and authentic network traffic data provide researchers with the opportunity to conduct thorough analyses, effectively train machine learning models, and benchmark the performance of intrusion detection systems (*Thakkar & Lohiya, 2020*).

## The NSL-KDD dataset

The NSL-KDD dataset contains records of the internet traffic seen by a simple intrusion detection network. The dataset contains 43 features per record, with 41 of the features referring to the traffic input itself and the last two are labels (whether it is a normal or attack) and score (the severity of the traffic input itself). Within the data set exists four different classes of attacks (*Chikkalwar & Garapati, 2025*): DoS, Probe, User to Root (U2R), and Remote to Local (R2L).

DoS attacks aim to slow down or prevent legitimate users from accessing a system by exhausting resources like bandwidth, CPU, and memory. U2R attacks involve an attacker escalating privileges from a normal user account to a root or administrator level, often through techniques like buffer overflow. R2L attacks occur when an attacker gains unauthorized access to a local system from a remote machine by exploiting vulnerabilities such as password cracking. Probing attacks, classified as passive, involve gathering information about a network or system to identify potential weaknesses before launching an active attack (*Elsayed, Mohamed & Madkour, 2024*).

## Evaluation

Machine learning evaluation involves assessing the performance and efficacy of a machine learning model. The assessment relies on diverse performance metrics, selected based on the specific objectives. In classification problems, metrics like accuracy, precision, recall,

F1-score, and area under the ROC curve (AUC-ROC) are widely employed (*Ahmed et al., 2024*). Accuracy measures the overall correctness of the model by calculating the proportion of correctly classified instances. Since IDSs need to be highly accurate in identifying normal and attack traffic. Precision represents the proportion of correctly identified attack instances among all predicted attack instances. High precision is crucial in IDS to minimize false alarms and avoid misclassifying normal traffic as attacks. Recall (Detection Rate) indicates the model's ability to detect all actual attacks. A high recall ensures that most attacks are detected, which is vital for security applications where missing an attack could have serious consequences. F1-score balances precision and recall, providing a more holistic measure of performance. This is particularly useful in imbalanced datasets, where focusing solely on accuracy can be misleading (*Tatachar, 2021*).

In a confusion matrix, True Positive (TP) and True Negative (TN) represent accurately predicted attack and normal records, respectively. False Positive (FN) and False Negative (FN) denote misclassified normal and attack records, respectively. This matrix provides a structured summary of the model's performance. From the indicators derived from the confusion matrix, key metrics include accuracy, detection rate, precision, and false alarm rate (FAR). Accuracy represents the ratio of correctly predicted records. Detection rate assesses the model's ability to predict all positive records accurately. Precision gauges the model's capability to avoid misclassifying negative records as positive. FAR quantifies the ratio of misclassifications for normal traffic.

$$\text{Accuracy} = \frac{TP - TN}{TP + TN + FP + FN}. \tag{6}$$

$$\text{Precision} = \frac{TP}{TP + FP}. \tag{7}$$

$$\text{Detection Rate} = \frac{TP}{TP + FN}. \tag{8}$$

$$\text{FAR} = \frac{FP}{FP + TP}. \tag{9}$$

## METHODOLOGY AND PROPOSED MODEL

This section describes the Optimized LSTM-RNNs IDS model as shown the proposed approach flowchart on Fig. 3. The LSTM-RNN architecture is employed to model the temporal dependencies in the sequential data, with LSTM cells retaining and propagating information across sequences. The model is trained using supervised learning, optimizing parameters through BackPropagation Through Time (BPTT). Fine-tuning and optimization techniques also applied to enhance the model's capabilities. Overall, this approach harnesses LSTM-RNNs and the CIC-IDS-2018 dataset to create a robust network IDS capable of accurately identifying network intrusions and enhancing network security.

The proposed IDS is built on a deeply structured deep learning pipeline that utilizes an optimized LSTM-RNN architecture. It is designed to address key shortcomings in traditional IDS models, ineffective learning from imbalanced datasets, and poor scalability
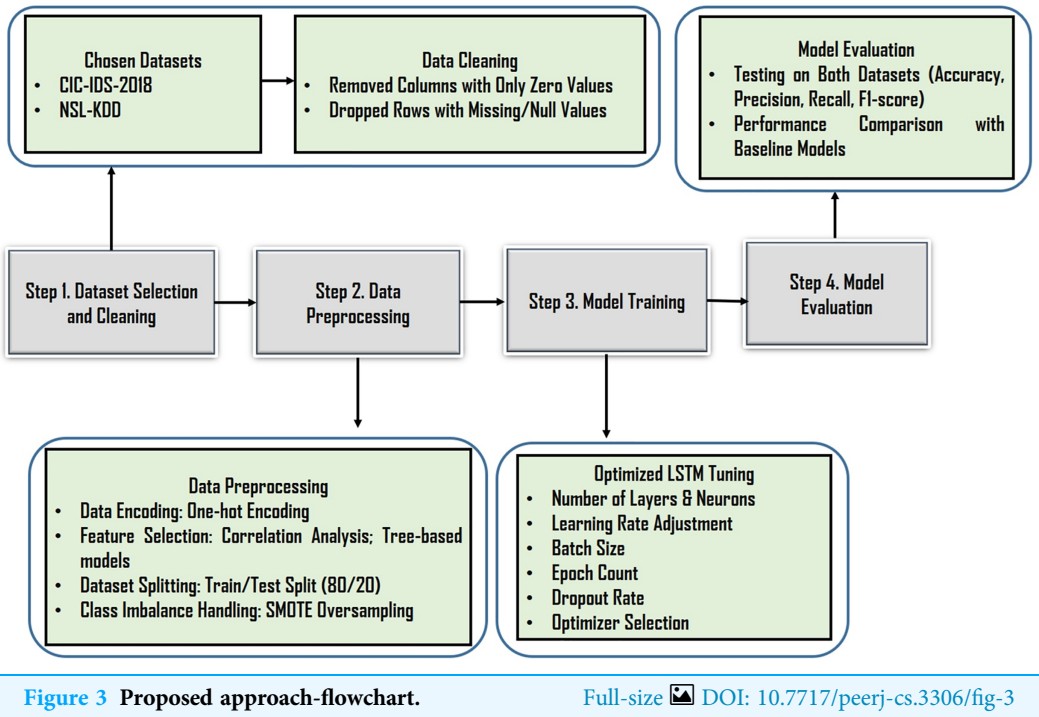

**Figure 3 Proposed approach-flowchart.**

for real-world network environments. The system's architecture, as illustrated in the model workflow diagram, demonstrates a comprehensive process integrating advanced data preprocessing, strategic data balancing techniques, and a finely tuned deep learning model, all aimed at delivering high-performance intrusion classification using the CIC-IDS-2018 and NSL-KDD datasets.

The initial phase emphasizes rigorous data cleaning and preparation The CIC-IDS-2018 and NSL-KDD datasets is thoroughly sanitized by removing zero-value columns that add no informational value and dropping rows with null values to maintain data integrity and prevent training inconsistencies. This cleaning significantly enhances the dataset's quality and relevance for subsequent modeling.

The preprocessing and transformation stage incorporates several technical innovations. First, categorical features such as network protocols are converted into machine-readable formats using one-hot encoding, avoiding false ordinal relationships. Second, data normalization is applied to ensure that numerical values fall within a uniform range, which is crucial for the efficient training of the LSTM model. Third, feature selection techniques primarily based on correlation analysis are used to eliminate redundant or irrelevant attributes. This not only improves model generalization but also reduces the risk of overfitting by allowing the model to learn from the most impactful features.

To combat class imbalance, an inherent challenge in intrusion detection tasks, the model employs the Synthetic Minority Over-sampling Technique (SMOTE). SMOTE generates samples for underrepresented attack types such as XSS, SQL Injection, and FTP-BruteForce, thus ensuring balanced training data, improved sensitivity to rare intrusions,

and reduced bias toward the dominant benign class. It is carefully applied after encoding and feature selection but before the training/testing split.

The core of the system is an optimized LSTM-RNN model explicitly designed to handle the sequential nature of network traffic data. It uses LSTM layers, allowing the model to learn complex hierarchical patterns over time especially useful for detecting multi-stage or evolving intrusions. Each layer consists of 80 hidden units, a number determined through hyperparameter tuning to balance model complexity and learning capacity. The training process utilizes backpropagation through time (BPTT), ensuring the model retains long-term dependencies while mitigating gradient vanishing problems. Additional optimization techniques, including tuning of learning rate, batch size, number of epochs, and the Adam optimizer, further refine the model's performance and stability.

The evaluation strategy adopts multiple metrics to assess the model comprehensively. Accuracy, precision, recall, and F1-score offer multi-dimensional insight into the model's effectiveness, especially in the context of imbalanced data. The FAR is also measured, providing a realistic assessment of the model's reliability in distinguishing normal from malicious traffic. Finally, visualization of training and validation loss and accuracy curves across epochs is used to monitor overfitting and ensure generalization to unseen data.

## Data preprocessing

Data preprocessing of the CIC-IDS-2018 dataset involves several key steps. Initially, the dataset is examined for missing values and duplicate entries, which can be addressed by removing or filling missing values and eliminating duplicates. Therefore, in this research, the dataset is examined for duplicate instances, and any duplicates found were removed to ensure data integrity (*Liu et al., 2022*). In the same manner missing values can disrupt the analysis and modeling process, the dataset was inspected for missing values and we removed all rows with missing values. Figure 4 depicts the cleaned CIC-IDS-2018 dataset distribution.

In the preprocessing of the CIC-IDS-2018 dataset, One-hot encoding was applied to the "protocol" feature, One-hot encoding transforms categorical data into a binary format to make it easier for machine learning algorithms to understand (*Hancock & Khoshgoftaar, 2020*). In the CIC-IDS-2018 dataset, applying one-hot encoding to the "protocol" feature would involve creating new binary columns for each unique protocol in the dataset. For instance, if the "protocol" feature had values like Tranmission Control Protocol (TCP), User Datagram Protocol (UDP), and Internet Control Message Protocol (ICMP), the one-hot encoding process would create three new columns, one for each protocol. For the "label" feature, we used ordinal encoding to assign numerical values based on the order or hierarchy of the labels. These encoding methods prepared the dataset for further analysis and modeling by converting categorical variables into suitable numerical representations.

The CIC-IDS-2018 datasets suffer from imbalanced class distributions, where normal instances greatly outnumber attack instances. This imbalance often biases models toward the majority class, leading to poor detection of rare but critical attacks. To address the issue of class imbalance in the CIC-IDS-2018 dataset, we applied oversampling specifically using the Synthetic Minority Over-sampling Technique (SMOTE). SMOTE generates synthetic

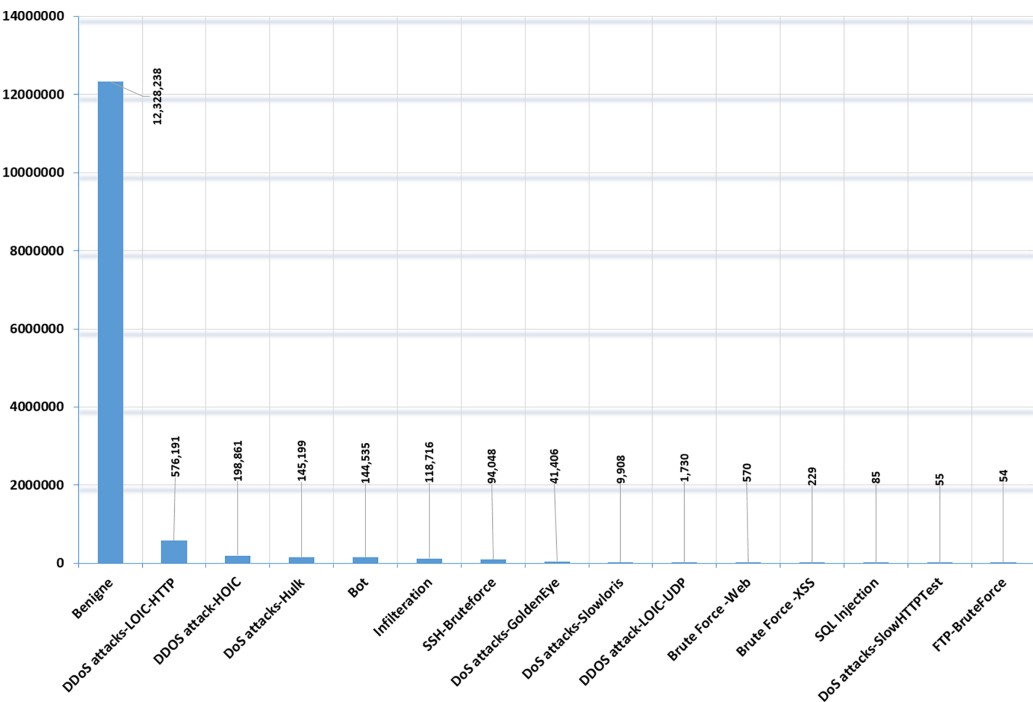

**Figure 4 Cleaned CIC-IDS-2018 dataset-benign and attack classes distribution.**

samples for the minority class by interpolating feature values between existing minority class samples. This technique helps to balance the class distribution and mitigate the impact of class imbalance on the modeling processes (*Zhang & Wang, 2023*). By oversampling the minority class using SMOTE, we increased the representation of the minority class which are DDOS attack-LOIC-UDP, Brute Force–Web, Brute Force–XSS, SQL Injection, DoS attacks-SlowHTTPTest and FTP-BruteForce. SMOTE improved the overall balance of the dataset. Applying SMOTE leads to more balanced and reliable model performance, particularly in detecting rare and critical attacks in NIDS. While it may introduce a minor computational overhead and overall fairness across classes typically makes it well worth applying in imbalanced datasets. Figure 5 shows the distribution of dataset after oversampling on randomly selected data to train the model.

## Experimental setup

Irrelevant or redundant features can increase model complexity and reduce performance. Feature selection techniques were applied to identify and remove such features. Methods such as correlation analysis or feature importance from tree-based models were utilized to select the most informative features for the analysis.

The preprocessed CIC-IDS-2018 dataset was split into training and testing data sets. A split ratio of 80–20 was employed, where 80% of the data was used for training, and 20% for testing. The training set was utilized for model training, allowing the model to learn patterns and relationships from the data. The testing set served as an independent dataset to evaluate the final performance of the optimized LSTM-RNN intrusion detection model.

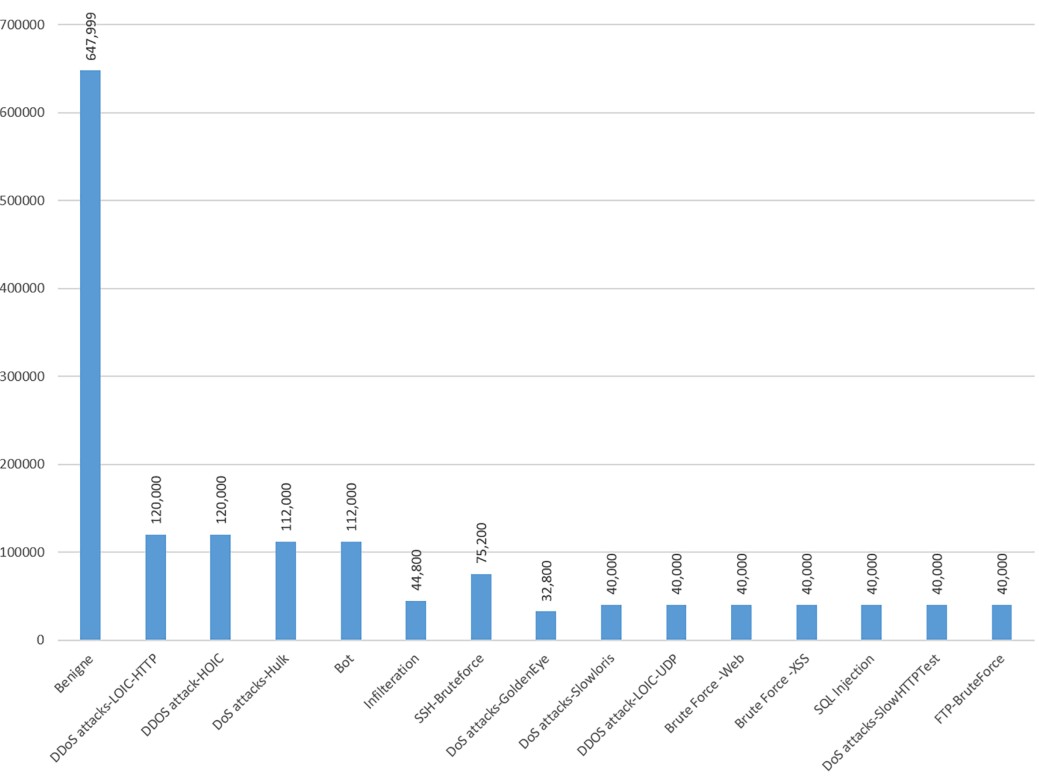

**Figure 5 Selected CIC-IDS-2018 dataset classes distribution after SMOTE.**

**Table 2 Hyper-parameters of experiments.**

| Dropout | Activation function | Learning rate | Optimizer | Epoch | Batch size | Loss function |
|---------|---------------------|---------------|-----------|-------|------------|---------------|
| 0.3 | Sigmoid | 0.01 | Adam | 300 | 32 | Binary cross entropy |

The LSTM-RNN architecture is optimized to capture the sequential nature of the network traffic data, which is crucial for intrusion detection. The number of LSTM layers is determined based on experimentation and empirical observations. A total of 80 hidden layer units within each LSTM layer is selected to balance the model's capacity and complexity. The optimal number of hidden units is determined through hyperparameter tuning. Hyperparameters, including learning rate, batch size, number of epochs, optimizer algorithm and loss function, were fine-tuned to optimize model performance through experiment. Table 2 shows the best hyperparameter selected in the experiment.

The optimized LSTM-RNN intrusion detection model is designed to effectively capture the temporal dependencies in the network traffic data and make accurate predictions for intrusion detection. Through the careful selection of feature subsets, appropriate dataset splitting, and architecture design, the model aimed to achieve high detection accuracy while minimizing false positives and false negatives.

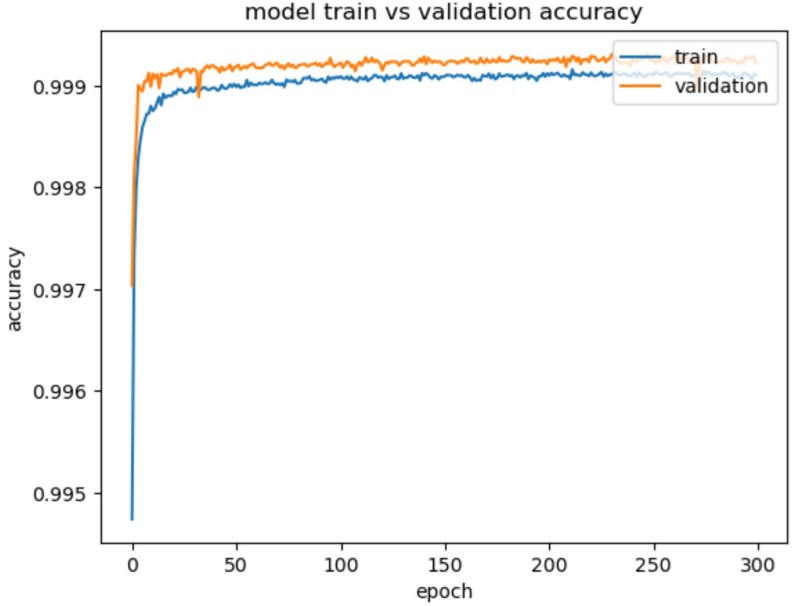

**Figure 6 Model train *vs.* validation accuracy for CIC-IDS2018.**

## RESULTS

Evaluation metrics like accuracy, precision, recall, and F1-score are used to assess the model's performance in detecting and classifying network intrusions. The experimental results provide a comprehensive analysis of the performance of the proposed Optimized LSTM-RNN based network IDS model. The model exhibits excellent accuracy on both the training and validation datasets for binary classification, achieving a training accuracy of 99.94% and a validation accuracy of 99.98%. The specific behavior of the accuracy curve in Fig. 6 shows that it is gradually increasing as the number of epochs increases. The increasing accuracy demonstrates the model's ability to correctly classify network traffic instances with a higher degree of precision. Our proposed model achieved a False Positive Rate (FPR)/FAR of 0.036%, confirming its ability to minimize false alarms.

The training loss, measured at 0.0016, signifies that the model effectively minimized errors during the training process. In addition, the validation loss, reported as 0.0011, is an essential metric that indicates the model's generalization ability. A low validation loss suggests that the model can effectively generalize its learnings to unseen data, demonstrating its robustness. The reported validation loss shows that the model achieves a high level of accuracy on the validation data, with minimal errors in its predictions. This highlights the model's capacity to handle new instances and unseen network traffic with a high level of accuracy and precision.

Figure 7, describes the training and validation loss *vs.* the number of epochs, shows consistent decrease in the validation loss throughout the 300 epochs. This indicates that the model steadily improves its performance over time. This suggests that the model is

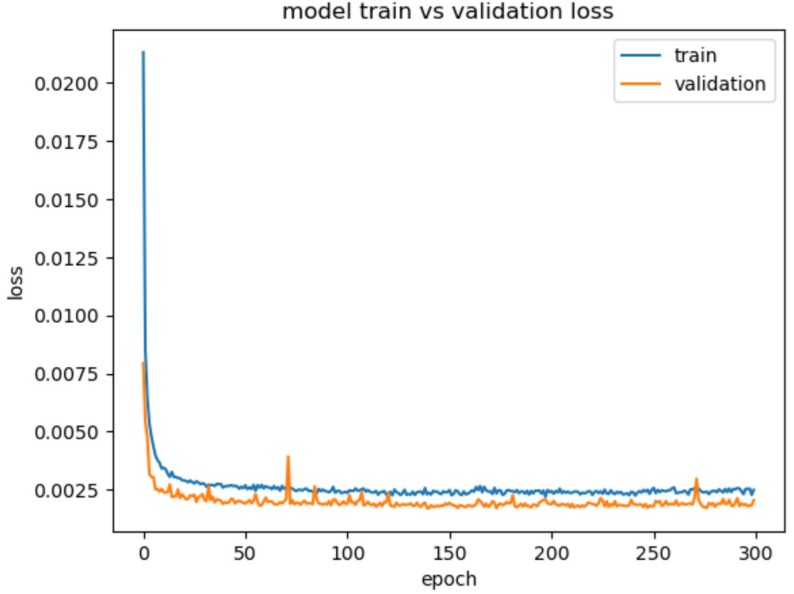

**Figure 7 Model train *vs*. validation loss for CIC-IDS2018.**

capable of capturing and learning intricate patterns and dependencies in the network traffic data. The gradual decrease in the validation loss demonstrates the model's ability to continuously refine its predictions and make increasingly accurate classifications.

Figures 8 and 9 illustrate the training and validation accuracy and loss curves of the proposed model on NSL-KDD dataset, respectively. In Fig. 8, both the training and validation accuracy curves show a consistent upward trend and gradually converge, indicating that the model is learning effectively without overfitting. The close alignment between training and validation accuracy suggests strong generalization to unseen data. Similarly, Fig. 9 shows a steady decrease in both training and validation loss, further supporting the model's stability during training.

Figures 10 and 11, show the confusion matrices for the CIC-IDS2018 and NSL-KDD datasets, respectively. These results demonstrate the high effectiveness of the proposed model in binary classification for network intrusion detection. For the CIC-IDS2018 dataset, the model achieved exceptionally low misclassification rates, with only five FPs and 21 FNs, indicating a strong ability to correctly identify both normal and attack traffic. Similarly, for the NSL-KDD dataset, the model recorded 31 FPs and 103 FNs, which still reflect solid performance. Overall, the low FP and FN counts across both datasets confirm the model's reliability in minimizing both missed attacks and false alarms.

Significant improvements in computational efficiency and execution time have been achieved through various model and system-level optimizations. LSTMs are powerful for capturing temporal dependencies in sequential network traffic data; however, their sequential nature can introduce latency during training and inference. To mitigate this,
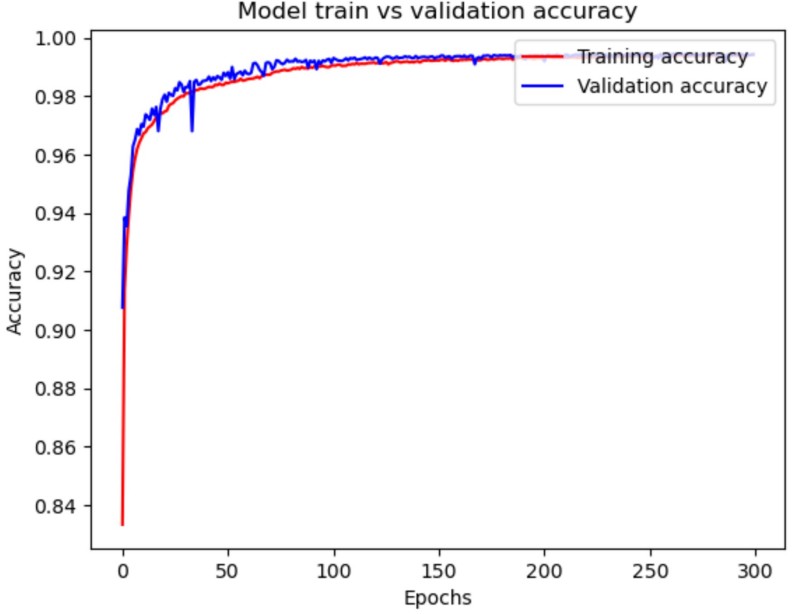

**Figure 8 Model train *vs*. validation accuracy for NSL-KDD.**

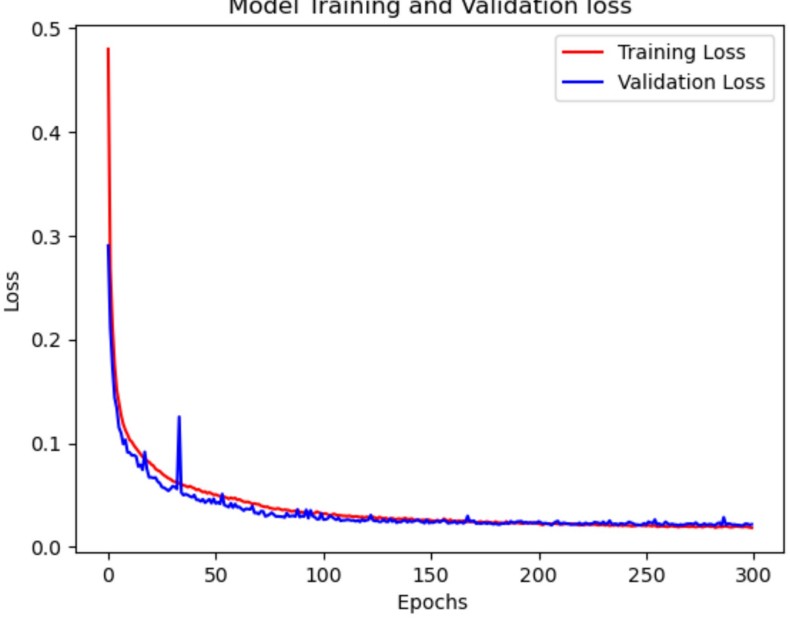

**Figure 9 Model train *vs*. validation loss for NSL-KDD.**

optimizations such as reducing the number of LSTM layers, limiting sequence length, and minimizing hidden unit sizes are done, and this decreases computational load without severely impacting accuracy.

|  | | Predicted Class | |
| --- | --- | --- | --- |
|  | | Attack | Normal |
| Actual | Attack | 30,326 | 5 |
|  | Normal | 21 | 13,860 |

**Figure 10 Confusion matrix for CIC-IDS2018 dataset.**

|  | | Predicted Class | |
| --- | --- | --- | --- |
|  | | Attack | Normal |
| Actual | Attack | 12,733 | 31 |
|  | Normal | 103 | 9,711 |

**Figure 11 Confusion matrix for NSL-KDD dataset.**

## DISCUSSION

The proposed LSTM-RNN model achieves the highest accuracy of 99.94% among all the compared models as shown in Table 3. It exhibits near-perfect precision and a high recall of 99.93%, indicating its ability to accurately detect network intrusions with an extremely low rate of false positives and false negatives. The F1-score of 99.95% further validates the model's good performance.

In addition to the CIC-IDS2018 dataset, the model's generalization was validated using the NSL-KDD dataset. The results remained consistently high, with 99.42% accuracy, 99.20% precision, 99.69% recall, and an F1-score of 99.45%. This cross-dataset validation confirms the model's robustness and adaptability in various intrusion detection scenarios.

The results highlight the superior performance of the proposed optimized LSTM-RNN model, which achieves the highest accuracy and precision scores among all the compared state-of-the-art models. The model demonstrates exceptional abilities in accurately classifying network intrusions while maintaining a balance between false positives and

**Table 3 Performance scores of the proposed model compared with related works.**

| Reference | Model | Datasets | Accuracy | Precision | Recall | F1-score |
|---|---|---|---|---|---|---|
| *Alzughaibi & El Khediri (2023)* | MLP-BP | CIC-IDS-2018 | 98.97% | 99.98% | 98.80% | 99.38% |
| *Bamber et al. (2025)* | CNN-LSTM | NSL-KDD | 95.00% | 99.61% | 89.00% | 94.00% |
| *Ullah et al. (2022)* | LSTM-GRU | CIC-IDS-2018 | 99.51% | 99.51% | 99.60% | 99.52% |
| *Rabih et al. (2025)* | LOF+CNN | CIC-IDS-2018 | 99.87% | 99.51% | 99.60% | 99.52% |
| *Balajee & Jayanthi Kannan (2023)* | PCA+SMO+FCM+AE | CIC-IDS-2018 | 95.5% | 94.7% | 47.8% | 63.5% |
| *Siddiqi & Pak (2022)* | MLP-BP | CIC-IDS-2018 | 97.75% | 97.97% | 97.76% | 97.86% |
| Proposed LSTM-RNN | LSTM-RNN | CIC-IDS-2018 | 99.94% | 99.98% | 99.93% | 99.95% |
| | | NSL-KDD | 99.42% | 99.20% | 99.69% | 99.45% |

false negatives. Notably, F1-score was corrected to 97.86% based on the reported precision and recall in *Siddiqi & Pak (2022)*. These findings emphasize the potential of the proposed LSTM-RNN model as a reliable and efficient solution for intrusion detection in computer networks.

In the case of an optimized RNN-based LSTM model for network intrusion detection, significant improvements in computational efficiency and execution time can be achieved through various model and system-level optimizations. LSTMs are powerful for capturing temporal dependencies in sequential network traffic data; however, their sequential nature can introduce latency during training and inference. To mitigate this, optimizations such as reducing the number of LSTM layers, limiting sequence length, and minimizing hidden unit sizes can help decrease computational load without severely impacting accuracy.

The optimized LSTM-RNNs based IDS provides substantial benefits and direct impacts for Information Technology (IT) users and organizations. The model ensures accurate detection and classification of network intrusions. In this research work the model is trained and tested with CIC-IDS-2018 dataset which has a substantial repository of network traffic, showcasing scalability and the capacity to manage extensive volumes of network data. These qualities enable real-time intrusion detection across diverse network environments, as evidenced by our results. By enhancing network security, the model contributes to safeguarding sensitive data and mitigating the risks of cyber-attacks. Additionally, the research paves the way for future advancements, such as multiclass classification and continuous retraining, further strengthening the capabilities of IDS systems. This research offers a reliable and efficient solution to enhance network security and protect against intrusions. The generalizability of the proposed LSTM-RNN based intrusion detection model is supported by the characteristics of the CIC-IDS-2018 dataset:

- This dataset emulates realistic enterprise-level network environments by capturing both benign and malicious traffic over an extended period, incorporating a wide range of protocols, ports, and user behaviors. This realistic simulation closely approximates real-world operational conditions, thereby facilitating the development of models trained on traffic patterns that are not artificially constructed or limited to synthetic scenarios.

- In addition, this dataset encompasses a comprehensive array of cyber-attacks including DoS, DDoS, brute force attempts, botnet activity, network infiltration, and web-based threats such as SQL injection and XSS. This extensive coverage of attack types enhances the model's exposure to diverse threat vectors, thereby improving its ability to detect not only known threats but also to identify anomalous patterns indicative of novel or evolving intrusions, a critical requirement for contemporary IDSs.
- Moreover, this dataset includes over 80 well-defined flow-based features for each traffic instance, providing a rich and diverse feature space for selection and engineering. This feature richness supports the model's ability to uncover intricate relationships between network characteristics and malicious behavior, thereby improving learning performance across a variety of attack scenarios and traffic conditions. Consequently, models trained on CIC-IDS-2018 are less likely to suffer from overfitting to narrow or overly simplistic datasets and are instead better positioned to generalize to real-world network environments.

The proposed optimized LSTM-RNN model improves the limitations of previous works by incorporating multiple stacked LSTM layers with 80 hidden units each, allowing the network to learn deeper and more complex temporal patterns. The architecture is further enhanced through fine-tuned training parameters such as learning rate, batch size, and loss function and preprocessing techniques like SMOTE for class balancing. This architectural depth and optimized design enable the proposed model to outperform others in detecting both simple and sophisticated intrusions.

This study makes both scientific and practical contributions to the field of cyber security, particularly in the area of network-based intrusion detection. Scientifically, it introduces an optimized LSTM-RNN architecture that effectively captures temporal dependencies in sequential network traffic data, providing a high-performing model for real-time intrusion detection. The model's robustness, validated through extensive evaluation on two benchmark datasets (CIC-IDS-2018 and NSL-KDD), highlights its potential for generalization and adaptability in dynamic threat environments. This work lays a foundation for future advancements in deep learning-based IDSs, including multiclass classification, enhanced model interpretability, and integration of continuous learning mechanisms. From a societal perspective, the proposed model contributes toward improving the reliability of network defense by enabling highly accurate and timely detection of malicious activity, as demonstrated through evaluation on realistic benchmark datasets. Although the model has not been tested in live enterprise or industrial environments, its performance indicates promising applicability for future deployment in operational settings. By reducing false alarms and enhancing detection accuracy, this work supports the development of more effective intrusion detection tools, helping organizations better detect, and respond to evolving cyber threats.

## CONCLUSIONS AND FUTURE WORK DIRECTIONS

The optimized LSTM-RNN-based network intrusion detection model demonstrated high performance on the CIC-IDS-2018 dataset. With an accuracy of 99.94%, precision of

99.98%, recall of 99.93%, and F1-score of 99.95%, the model showcased its ability to effectively distinguish between normal and malicious network traffic. The results indicate its potential as a reliable tool for detecting and mitigating network intrusions, thereby enhancing the security of network environments. Furthermore, the promising outcomes lay the foundation for future work in the field, such as expanding the model to handle multi-class classification, improving interpretability, and exploring continuous retraining and adaptation mechanisms.

The achievements of this study highlight the significance of deep learning techniques, particularly LSTM-RNN, in the domain of network intrusion detection. The model's high accuracy and precision reflect its ability to accurately classify instances, while the good recall and F1-score demonstrate its capability to effectively identify malicious traffic, minimizing the risk of false negatives. By leveraging the power of deep learning, this research contributes to the ongoing efforts in building robust and efficient network security systems.

In future works, extending the proposed LSTM-RNN model to handle multiclass classification for network-based intrusion detection would involve developing techniques to accurately classify different types of network intrusions, such as DDoS attacks, SQL injection and malware. Expanding the model's capabilities to handle a broader range of intrusions would enhance its usefulness. Finally, a potential direction for future research is the exploration of attention-based architectures, such as Transformers, which are increasingly used for modeling complex sequential dependencies. This would enable a comparative assessment against the optimized RNN and LSTM approach developed in this study for intrusion detection.

## ACKNOWLEDGEMENTS

The authors would like to acknowledge the NURTURE Partnership Project for providing the collaborative platform that facilitated this research. Special thanks are extended to Dr. Alemu Moges Belay for his constructive input and engagement throughout the study.

### Funding

This study is supported _via_ funding from Prince Sattam bin Abdulaziz University project number (PSAU/2025/R/1446). The funders had no role in study design, data collection and analysis, decision to publish, or preparation of the manuscript.

### Grant Disclosures

The following grant information was disclosed by the authors:
Prince Sattam bin Abdulaziz University: PSAU/2025/R/1446.

### Competing Interests

The authors declare that they have no competing interests.

## Author Contributions

- Abraham Berhanu conceived and designed the experiments, performed the experiments, analyzed the data, performed the computation work, prepared figures and/or tables, authored or reviewed drafts of the article, and approved the final draft.
- Sabarathinam Chockalingam conceived and designed the experiments, performed the experiments, analyzed the data, performed the computation work, prepared figures and/or tables, authored or reviewed drafts of the article, and approved the final draft.
- Jemal Abawajy conceived and designed the experiments, performed the experiments, analyzed the data, performed the computation work, prepared figures and/or tables, authored or reviewed drafts of the article, and approved the final draft.
- Shegaw Anagaw Mengiste conceived and designed the experiments, performed the experiments, analyzed the data, performed the computation work, prepared figures and/or tables, authored or reviewed drafts of the article, and approved the final draft.
- Shabbab Ali Algamdi performed the experiments, analyzed the data, performed the computation work, prepared figures and/or tables, authored or reviewed drafts of the article, and approved the final draft.
- Dereje Ashenafi conceived and designed the experiments, performed the experiments, analyzed the data, performed the computation work, prepared figures and/or tables, authored or reviewed drafts of the article, and approved the final draft.

## Data Availability

The CIC-IDS-2018 Dataset is available at: https://www.unb.ca/cic/datasets/ids-2018.html.

The NSL-KDD Dataset is available at: https://web.archive.org/web/20150205070216/http://nsl.cs.unb.ca/NSL-KDD.

## Supplemental Information

Supplemental information for this article can be found online at http://dx.doi.org/10.7717/peerj-cs.3306#supplemental-information.

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
