# Peer review of "A recurrent neural network for network-based intrusion detection"

_PeerJ Computer Science, doi:10.7717/peerj-cs.3306_

## Round 0.1 · original submission · Major Revisions

· Academic Editor

Major Revisions

Reviewers have now commented on your paper. The reviewers have raised concerns regarding contribution, English grammar, experimental setup, results, and comparisons with previously reported approaches. These issues require a major revision. Please refer to the reviewers’ comments at the end of this letter; you will see that they advise you to revise your manuscript. If you are prepared to undertake the work required, I would be pleased to reconsider my decision. Please submit a list of changes or a rebuttal against each concern when you submit your revised manuscript.

Thank you for considering PeerJ Computer Science for the publication of your research.

With kind regards,

**Language Note:** The review process has identified that the English language must be improved. PeerJ can provide language editing services - please contact us at [email protected] for pricing (be sure to provide your manuscript number and title). Alternatively, you should make your own arrangements to improve the language quality and provide details in your response letter. – PeerJ Staff

·

Basic reporting

-The language used is clear.
-There are some grammatical issues and inconsistencies in technical terminology.
-Figure 3, 6, 7 need improvements and the confusion matrix visualization is missing.

Experimental design

The class imbalance issue is addressed using SMOTE, but a more detailed discussion of its impact on model performance would be valuable.

Validity of the findings

-The evaluation is limited to binary classification (normal vs. attack) rather than multi-class classification of specific attack types.
-There is no discussion of computational efficiency or execution time.
-It lacks detailed analysis of false positives and false negatives

Additional comments

-Abstract need to be adjusted to be more logical with good structure.
-This manuscript include comparisons with existing approaches.
-Need more detailed explanation of how the proposed LSTM-RNN architecture differs from existing approaches.

Reviewer 2 ·

Basic reporting

The impact of the paper must be written clearly. The discussion part must be added in order to discuss the results and comparisons with the other recently proposed methods. The paper should be carefully revised by a native English speaker or a professional language editing service to improve the grammar and readability. The more recent studies must be added.

Experimental design

The different datasets must be also be used to understand the performance of the proposed approach. The classical techniques are applied such as RNN and LSTM. the other attention based techniques can also be applied. The method must also applied on the multiclass classification

Validity of the findings

The performance of the proposed method must also be compared with the recently proposed deep learning strategies and approaches.

Cite this review as

Reviewer 3 ·

Basic reporting

The article has clear language in most places. However, some grammatical errors can still be found in it. For example, in the abstract (line 16), “detect and classify” should be “detects and classifies”. Another example is line 110, where “focuses” should be “focus”. A careful check for the language should be done throughout the article to address these issues.

The article uses the CIC-IDS-2018 dataset but does not correctly cite the dataset in the bibliography as required by its publisher (https://www.unb.ca/cic/datasets/ids-2018.html).

The article claims to propose an optimized LSTM-RNN model. More information is necessary to support the claim that the model is optimized compared to stock LSTM models, which is not clear in the article.

Experimental design

Only one dataset is used in the experiment. In table 1, all related works have at least two datasets for their experiments. More experiments on these datasets is necessary to show the generalizability of the article's approach.

Validity of the findings

no comment

Cite this review as

·

Basic reporting

1. The language used throughout the paper is professional and easy to follow, which enhances readability and comprehension. The article is structured professionally, with appropriate figures and tables included.
2. The overview in the proposed approach flowchart (Figure 3) could benefit from additional details about each step. This would allow readers to quickly grasp the major works, innovations, and techniques employed in the paper.
3. The results presented in Table 3 require validation, as there are discrepancies in the F1 scores relative to their corresponding precision and recall values. The author should ensure that all data is accurate and clearly defined.

Experimental design

1. The research presents original primary work relevant to the journal's Aims and Scope. The research question is defined, but the novelty of the proposed LSTM-RNN model and the data processing/enhancing methods is questionable. While the author claims to have made improvements to the model structure, a clearer justification of these enhancements in comparison to existing models, such as Ullah's 2022 LSTM-GRU, Hnamte's 2023 LSTM-AE, Han and Pak's 2023 Hierarchical LSTM, is necessary.
2. The evaluation of the proposed model includes baseline models from other papers, which is commendable. However, the reliance on published results raises concerns about the comparability of model performances due to potential differences in preprocessing methods and data augmentation processes. A more rigorous approach would involve testing the models on the same inputs for evaluation baselines.

Validity of the findings

The impact and novelty of the findings are not thoroughly assessed. While the work is generally acceptable, it lacks significant novelty in the proposed model and its application to the network intrusion detection problem.

Additional comments

Overall, the work is satisfactory, but it would benefit from a more in-depth exploration of the model's novelty and improvements over existing approaches.

---

## Round 0.2 · Minor Revisions

· Academic Editor

Minor Revisions

All concerns raised by the reviewers have been addressed satisfactorily; however, the manuscript requires further work on the experimental setup. This issue requires a minor revision. If you are prepared to undertake the work required, I would be pleased to reconsider my decision. Please submit a list of changes or a rebuttal against each point that is being raised when you submit your revised manuscript.

·

Basic reporting

Authors are addressed all comments nicely.

Experimental design

Authors are addressed all comments nicely.

Validity of the findings

Authors are addressed all comments nicely.

Additional comments

Authors are addressed all comments nicely.

·

Basic reporting

- The manuscript is readable and well structured; figures and tables are appropriately placed and support the narrative.
- Background and literature context are adequate. The author clarifies the main contribution and situates the work within prior studies.
- The addition of a second benchmark dataset strengthens results. Figures and tables (except one noted data inconsistency) provide required information.

Experimental design

- The study presents primary experimental work relevant to network intrusion detection.
- The author frames the gap as high false alarm rates, scalability and generalization. The main contributions and how they address this gap are better articulated in the revision.

Validity of the findings

- The revision makes the contribution clearer, but the methodological novelty relative to existing LSTM-based variants remains incremental. The manuscript now argues practical value (improved benchmarking on an additional dataset), which increases impact, though theoretical novelty is modest.
- Results are reported across datasets; one data inconsistency undermines confidence (see below). The manuscript omits False Positive Rate (FPR) or related metrics; given the stated research gap of high false alarms, reporting FPR would strengthen evaluation.

Additional comments

A few minor corrections:
1. Correct the F1 inconsistency in Table 3 e.g., Siddiqi and Pak’s row shows precision 0.9797 and recall 0.9776 but an F1 of 0.9774, which is mathematically inconsistent. Verify and correct all computed metrics.
2. Report False Positive Rate (or equivalent) if available; if not, add a brief statement acknowledging that FPR was not evaluated and avoid suggesting false-alarm improvements.

---

## Round 0.3 · accepted · Accept

· Academic Editor

Accept

I am pleased to inform you that your work has now been accepted for publication in PeerJ Computer Science.

Thank you for submitting your work to this journal. I look forward to your continued contributions on behalf of the PeerJ Computer Science editors.

With kind regards,

·

Basic reporting

The author has addressed all my comments, and I have no further concerns.

Experimental design

The author has addressed all my comments, and I have no further concerns.

Validity of the findings

The author has addressed all my comments, and I have no further concerns.

Additional comments

The author has addressed all my comments, and I have no further concerns.